# From Black Box to Algorithmic Insight: Explainable AI in Graph Neural Networks for Graph Coloring

**Elad Shoham [1], Havana Rika [2], Dan Vilenchik [1]**

[1] Ben-Gurion University of the Negev
[2] The Academic College of Tel Aviv-Yaffo
shellad@post.bgu.ac.il, havanari@mta.ac.il, vilenchi@bgu.ac.il

## Abstract

Despite advances in neural networks for solving combinatorial optimization problems using Graph Neural Networks (GNNs), understanding their learning processes and utilizing acquired knowledge remains elusive, particularly in imperfect models addressing NP-complete problems. This gap underscores the need for Explainable AI (XAI) methodologies. In this study, we undertake the task of elucidating the mechanisms of a specific model named GNN-GCP trained to solve the Graph Coloring Problem (GCP). Our findings reveal that the concepts that underpin the operation of GNN-GCP resemble those of hand-crafted combinatorial optimization heuristics. One prominent example is the concept of "support of vertex $v$ with respect to a given coloring of the graph", which is the number of neighbors that $v$ has in each color class other than its own. By providing insights into the inner workings of GNN-GCP, we contribute to the larger goal of making AI models more interpretable and trustworthy, even in complex settings such as combinatorial optimization problems.

## 1 Introduction

AI systems regularly match or exceed human performance in complex computational tasks, yet their decision-making processes often remain opaque. Understanding these processes is crucial for building trust, making explainable AI (XAI) an essential field of research.

While traditional XAI methods like SHAP (Lundberg and Lee 2017), LIME (Ribeiro, Singh, and Guestrin 2016), and GradCAM (Selvaraju et al. 2017) focus on feature attribution, they struggle to capture higher-level patterns in complex domains such as combinatorial optimization. Take, for example, the graph coloring problems where one is asked to color the vertices of a graph $G$ with the minimal number of colors so that no two vertices of the same color share an edge (we shorten the problem as GCP or $k$-coloring, where $k$ is the maximum number of allowed colors). Instead of merely highlighting important vertices or edges, we would like to identify algorithmic concepts that explain how the ML pipeline finds, if it does, a legal $k$-coloring of the graph.

Our research embraces concept-based cognition, where complex decision-making is understood through interpretable concepts represented in the neural network's latent

space (Guo et al. 2024). These concepts must be knowledge-bearing, transferable to both humans and machines and minimal in their representation.

Graph coloring presents unique challenges beyond traditional concept learning. A Graph Neural Network (GNN) solves the coloring problems through iterative message-passing that maintains and updates a partial coloring solution (Lemos et al. 2019; Wang, Yan, and Jin 2024; Ijaz et al. 2022a; Schuetz et al. 2022; Li et al. 2022; Colantonio et al. 2024). This creates a dynamic interplay between the input graph structure and the evolving solution state—a scenario fundamentally different from standard classification tasks, such as image processing or sentiment detection in text.

Our analysis focuses on one specific pre-trained model, GNN-GCP (Lemos et al. 2019), due to its unique architectural simplicity and unconstrained learning approach. The model consists of basic message-passing layers followed by an MLP, with its only training objective being to predict whether a graph is 3-colorable through binary cross-entropy loss (See section 4). Unlike other approaches that incorporate domain-specific heuristics, architectural constraints, or guidance through the loss function, GNN-GCP's minimal structure and open-ended learning objective provide an ideal setting to study what strategies naturally emerge. This unconstrained learning environment makes it particularly interesting from an explainable AI perspective—understanding how a model arrives at effective solutions without explicit guidance.

We extend concept learning to address these challenges in graph coloring, examining how concepts depend on both static graph structure and dynamic solution states. We find that the GNN tracks combinatorial properties like vertex degree (static) and potential color conflicts with respect to the current coloring solution (dynamic), all encoded in a unique spatial geometry that the embedding induces.

By analyzing how GNNs naturally develop problem-solving strategies, we gain insights into both neural network behavior and the fundamental structure of graph coloring problems.

In this work, we discover the concepts that the GNN learned during training and applied during test time to solve colorability. These concepts fully manifest in successful executions, where a legal coloring is achieved using the GNN, while in failed executions, some of them remain muted.

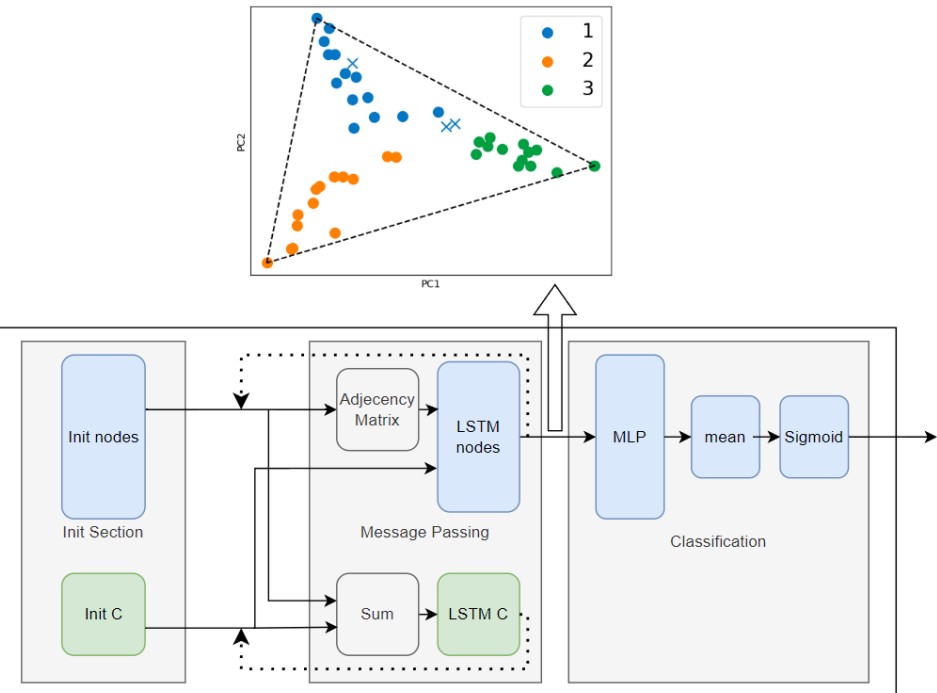

Figure 1: GNN-GCP model architecture (Lemos et al. 2019) and its 2D embedding visualization. The model updates vertex embeddings through three inputs: the vertex's previous state, neighbor embeddings (via adjacency matrix), and global graph context (C). The vertex-LSTM outputs are projected using PCA to a 2D space, revealing a triangular configuration where each color class is arranged in a different third of the triangle. Some vertices are closer to the triangle vertices, and some are to the center of mass. This arrangement is driven and explained by algorithmic concepts that we reveal which underpin the working of the GNN.

While we explored a wide range of potential concepts, we focus here on the learned and successful ones that emerged as most relevant to the canonical $k = 3$ case. Figure 1 illustrates our findings.

We urge the reader to note that as they are reading the concepts, they are actually reading a description of the GNN as a white-box textbook algorithm.

- GNN-GCP encodes graph coloring concepts in the vertex embedding, which are updated over the GNN iterations via a message passing exchange along the graph edges. The concepts are expressed in a geometric manner. The hallmark of this geometry is that the 2D PCA projection of the embedding forms a triangular shape. A 3-color assignment can be derived from the embedding by mapping graph vertices to their nearest triangle vertex.

- We introduce $d_v^{(t)}$, the average distance between vertex $v$'s embedding and its graph neighbors' embeddings in iteration $t$. The average value of $d_v^{(t)}$, averaged over the vertex set $V$, increases over the iterations, driving the vertices' embeddings towards the triangle's vertices.

- We discovered that the distance $d_v^{(t)}$ serves as a confidence level in $v$'s current coloring, where larger values of $d_v^{(t)}$ indicate higher confidence. We further found that the concept underpinning the confidence is that of "support"

(see Section 2.1), which reflects how many neighbors a vertex has in each color class. A key insight emerges: a vertex's confidence increases proportionally with an increase in its support – driving high-confidence and high-support vertices to the triangle's vertices. As a result, high-confidence vertices are less likely to be grouped with their neighbors in the same color class, reducing the probability of monochromatic edges and serving as anchors for the emerging coloring. This geometric positioning induces the different color classes, with the triangle's vertices acting as color class representatives.

Thus, through the concept of confidence and support, encoded geometrically in the GNN's embedding, we could explain how the model learns and applies key algorithmic. These concepts were actually used in the past in the design of hand-crafted algorithms for graph coloring, such as (Alon and Kahale 1994).

## 2 Graph coloring

Graph coloring, one of the fundamental problems in graph theory, asks to assign colors to vertices such that no adjacent vertices share the same color (Karp 2010a; Lewis 2016). More formally, given an undirected graph $G = (V, E)$ we seek a mapping $c : V \rightarrow \{1, \ldots, k\}$ such that $(u, v) \in E \implies c(u) \neq c(v)$. While this definition is straight-

forward, the problem exhibits rich structural properties that have inspired various solution approaches and is one of the first problems shown to be NP-complete (Karp 2010b).

The interplay between local and global properties of graphs helps illustrate why the graph coloring problem is NP-complete. Dense subgraphs often demand more colors to ensure no adjacent vertices share the same color, whereas sparse regions allow greater flexibility in color assignments. This complex relationship between local structure and global colorability makes it difficult to devise efficient algorithms that work universally. Understanding this tension is key to developing effective heuristics and highlights the inherent computational challenges of the problem.

Nevertheless, the study of graph coloring has led to the development of several fundamental heuristics that underpin modern algorithmic approaches. At the heart of any coloring solution lies the notion of color classes — independent sets of vertices that can share the same color. The chromatic number $\chi(G)$ is the minimum number of colors needed for a valid coloring. Heuristic algorithms often employ sequential coloring strategies, progressively assigning colors to vertices while maintaining validity constraints. Some examples are greedy coloring (color vertices in order of smallest available colors) or saturation degree (number of differently colored adjacent vertices). When conflicts arise, color exchange operations — where colors are swapped between vertices —provide a mechanism for resolving these conflicts while preserving the validity of the overall coloring. The greedy algorithm was rigorously analyzed by (Krivelevich 2002) for random graphs, establishing the set of parameters for which this algorithm succeeds.

Through this paper, we only consider the 3-coloring problem, as this is the domain of the GNN that we study.

## 2.1 Support in graph coloring

In graph coloring, a *conflict* occurs when two adjacent vertices share the same color.

The notion of *support*, first introduced in (Alon and Kahale 1994), provides an intuitive measure of the confidence in the current color assignment $C(v)$ of a vertex $v$, with respect to a (not necessarily legal) 3-coloring $C$. To define it, consider $v$ and its neighbors: count how many of its neighbors belong to each of the other two color classes under $C$, and take the minimum of these two counts. This value, called the support of $v$, reflects how "entangled" $v$ is with the two alternate color classes.

The support captures how many new conflicts would arise if $v$ were reassigned from its current color $C(v)$ to any other color. Specifically, if the support is $s$, then flipping the color of $v$ introduces $s$ new conflicts, which would require flipping the colors of at least $s$ other vertices to resolve. Thus, a high support value $s$ suggests two possibilities:

1. $v$ is likely correctly assigned under $C$, as changing $C(v)$ would disrupt the coloring significantly. 2. If $v$ is incorrectly assigned, then $C$ is incorrect on at least $s$ additional vertices.

When $C$ is close to a legal coloring, large support values imply a higher likelihood that $v$ is correctly colored, reinforcing the stability of the coloring near $C$. This makes support a valuable metric for assessing the reliability of $C(v)$

in iterative or heuristic graph coloring algorithms such as (Alon and Kahale 1994; Bui et al. 2008; Dupin 2024; Wu, Luo, and Su 2013; Douiri and Elbernoussi 2015).

## 3 Related work

### 3.1 Concept Learning

Concept-based methods have emerged as a powerful approach to model interpretability, offering a higher level of abstraction compared to traditional feature and data-centric interpretability methods (Sundararajan, Taly, and Yan 2017; Lundberg and Lee 2017; Ribeiro, Singh, and Guestrin 2016; Schut et al. 2023). These methods aim to provide model explanations that are more intuitive and informative for human practitioners. Research in this field has branched into several distinct approaches. Supervised concept excavation (Achtibat et al. 2022; Melis and Jaakkola 2018; Schut et al. 2023; Kim et al. 2018; Bau et al. 2017) relies on labeled data to identify and validate concepts. In contrast, unsupervised concept mining (Yeh et al. 2020; Ghandeharioun et al. 2021; Ghorbani et al. 2019) seeks to discover inherent concepts without labeled data, learning directly from the structure of problem instances.

A closely related paradigm, known as "Mechanistic Interpretability" or "Mechanistic Design," takes inspiration from reverse engineering compiled binary code. This approach aims to deconstruct neural networks to identify specific functionalities by examining individual components of the model (Olah et al. 2022; Michaud et al. 2023; Wang et al. 2023; Olah et al. 2020; Nanda et al. 2023). Recent work has extended concept learning to Graph Neural Networks (GNNs) (Rakaraddi et al. 2022; Sun, Li, and Zhang 2022; Gonzalez, Holder, and Cook 2002), focusing on identifying patterns in input data for structural learning and prerequisite information.

Our work crucially departs from previous concept learning methods in our considered domain. While previous works assume that the concepts are imbued in the data, we consider our dataset as a set of Turing machines (in our case, a Turing machine for solving graph coloring). In this case, the concept depends on the machine's transition function as well as the working memory. This raises the level of complexity compared to finding concepts in textual data, images, or even in games, as the network must learn to recognize and utilize patterns in both the graph structure and its own solution process. (Shoham et al. 2024) is the work closest to ours, where they use similar analysis and techniques for the SAT problem..

### 3.2 Neural networks for graph coloring

Graph coloring has a rich history of algorithmic developments of classical heuristics like (Wu, Luo, and Su 2013; Dupin 2024; Ijaz et al. 2022b; Karger, Motwani, and Sudan 1998), which are fundamental bases for solving the GCP.

Neural network-based approaches have recently emerged as powerful tools for tackling graph coloring challenges. End-to-end neural architectures (Lemos et al. 2019; Wang, Yan, and Jin 2024; Ijaz et al. 2022a; Schuetz et al. 2022;

Li et al. 2022; Colantonio et al. 2024) have shown promising results in learning to solve the problem using a message passing GNN, some use supervised data and some unsupervised which use a specific loss function fitting the problem.

Another emerging approach seeks to develop generalized models capable of solving diverse combinatorial optimization problems (Boisvert, Verhaeghe, and Cappart 2024; Marty et al. 2023; Cappart et al. 2023), including max-cut and graph coloring. Researchers aim to create universal neural network architectures that can effectively address multiple optimization challenges through a unified approach. (Yau et al. 2023) even suggested that GNNs could potentially serve as optimal approximation algorithms across different combinatorial problems.

## 4 Model

GNN-GCP was introduced by (Lemos et al. 2019) and is an end-to-end message-passing GNN model designed to solve the graph coloring problem. The architecture of the model, drawn in Figure 1, is relatively simple; it has a global state component $C$, which is an LSTM with a hidden state of dimension $d = 64$, taking the previous global state and the sum of all vertices' hidden states. The second component of the GNN is an LSTM with one hidden state and a $1 \times d$ input matrix, applied in parallel (broadcasting) to all $n$ vertices. The application for each vertex is performed over the global state and the sum of the neighboring vertices' embedding from the previous iteration (that's where the adjacency matrix plays a role). The third component is a $1 \times d \Rightarrow 1 \times 1$ MLP that is applied to the $n \times d$ vertices' embeddings matrix (in parallel) and outputs a $n \times 1$ vector. The entries of the vector are averaged and used to predict, using a sigmoid gate, whether the given graph is 3-colorable or not. The model is trained with the cross-entropy loss, meaning the only input data was the adjacency matrix and the single-bit whether the instance is 3 colorable or not.

The training data used in (Lemos et al. 2019) are random graph instances near the theoretical threshold of 3-colorability. These are considered hard combinatorial instances (Zdeborová and Krzakala 2007; Bapst et al. 2014). The instances were generated by staring from an empty graph, and each time adding a random edge, until (using an exact solver to verify) the graph becomes not 3-colorable. The pair of graphs, the 3-colorable and the not-3-colorable, differing on a single edge, was added to the train set. A total of 4096 instances were used for training. We received the trained model from the authors of the paper.

(Lemos et al. 2019) report that the model achieved an accuracy between 75% - 82% with 40 to 60 vertices and 32 iterations of message passing. We verified this claim by running the model several times (usually three) for 32 iterations and considering a success if one of the runs returns a truth value for colorable instances and if all runs return a false value for non-colorable instances. This method was inspired by the original authors in (Lemos et al. 2019). However, we also noticed that increasing the number of vertices to 1000, changing the density of instances, or running another number of iterations, decreases the accuracy drastically. This can be attributed to the very small training set; for comparison,

a similar GNN for solving SAT was trained on over 1M instances (Selsam et al. 2019).

To extract a 3-coloring from the embedding, we ran the $k$-means algorithm (with $k = 3$) on the embedding of the last iteration and assigned each cluster with a different color. We found that in only about 5% of 3-colorable instances, a legal coloring was found. The way we ran the model was to inspect at each iteration whether the coloring was valid and stop when the answer was yes, or if a cap of 150 iterations was reached.

Let us note that even though the model is not robust to out-of-distribution data, and the ability to find the actual coloring is quite low, we still were able to uncover the algorithmic concepts underpinning the model's operation when it was right, showcasing the potential of such XAI methods even on non-perfect models.

## 5 Data

Our experimental evaluation employs a diverse set of graphs that span different sizes and density regimes, allowing us to analyze our algorithm's behavior across varying problem difficulty levels, both colorable instances and non-colorable instances. The dataset comprises both random graphs and planted solution instances. Let us stress that the data that we describe was only used to test the trained GNN that was shared with us by the authors of (Lemos et al. 2019).

### 5.1 Random graph instances

We generate random graph instances across multiple scales, with vertex counts $n \in \{45, 100, 500, 1000\}$. For each graph size, we consider different constraint densities parameterized by $c \in \{1, 2, 3, 3.5, 4, 4.5\}$, where $c$ represents the ratio of constraints (edges) to variables (vertices). Specifically, for a graph with $n$ vertices, we include $cn$ edges randomly distributed among possible vertex pairs.

The chosen range of $c$ values is particularly significant as it is below the phase transition phenomenon in graph coloring. The phase transition marks a critical threshold where problems typically transition from being colorable to uncolorable (Bapst et al. 2016; Zdeborova and Krzakala 2007). For 3-coloring random graphs, theoretical and empirical studies have shown this transition occurs around $c \approx 4.69$. Our range of $c$ values deliberately stays below but reaches close to this critical threshold to make sure the instances are colorable, as the concepts that we are interested in are algorithmic concepts of colorability. A tangent algorithmic problem is certifying non-colorability (or refutation). This is a much harder problem as it lies in CO-NP and is beyond the scope of this work.

### 5.2 Planted Solutions

Random 3-colorable instances have an inherent degree limit. The planted distribution allows the generation of graphs with an arbitrary average degree by the following procedure: First, partitioning the $n$ vertices into $k = 3$ color classes of size $n/k$. Then, include each edge $(u, v)$ that connects vertices $u, v$ in different color classes with probability $p$, independently for every pair. The average degree in a planted graph is $2pn/3$.

We tested the following parameters set: $p \in \{0.3, 0.5, 0.8\}$ and $n = 45, 100, 500, 1000$.

# 6   Experiments and results

In this section, we explain in detail how GNN-GCP attempts to solve the graph coloring problem and which algorithmic concepts it uses. At the end of the section, the reader should be able to write a simple white-box description of the GNN.

To run the experiments, we used a personal computer with an Nvidia RTX 3080, 370 GPU, 64GB RAM, and an AMD Ryzen 9 5650x CPU. The code can be found at https://github.com/HavanaLab/graph_coloring. We obtained the weights of GNN-GCP from the original authors of (Lemos et al. 2019) and translated them from TensorFlow to Torch.

The original paper of (Lemos et al. 2019) ran the model for 32 message-passing iterations before classifying the 3-colorability of the graph. Then, to extract the color assignment, the last message-passing layer's embedding was taken, and the $k$-mean clustering algorithm was used to assign a color to each vertex. We ran it for 150 iterations, checking at each iteration if the 3-coloring is legal, stopping at the first iteration where a legal coloring was found, or when the 150 cap was reached.

Our test set included two datasets $D_S$ and $D_F$ ('S' for success and 'F' for failure), constructed as follows: pick a random value of $n, c$ (from the set of values described in Section 5), generate a random graph $G$ and run GNN-GCP on it. If a legal 3-coloring was found, add $G$ to $D_S$; otherwise, add $G$ to $D_F$. Stop when each is of size 50. Repeat this procedure with planted instances, choosing $n$ and $p$ at random, and stop when the size of $D_S$ and $D_F$ reached 100.

## 6.1   Geometric shape of embedding

We perform a two-dimensional Principal Component Analysis (PCA) on the model's message-passing embedding to gain a geometric understanding of the embedding space. We denote the first principal component as PC1 and the second as PC2. PCA was computed over the $d \times d$ (recall $d = 64$) covariance matrix of the embedding per instance per iteration.

We identified a recurring 2D geometric configuration in successful executions (those leading to a legal 3-coloring) characterized by "three legs" or "branches" emerging from a single center of mass. These legs are most compactly bounded by a triangular shape. This triangular structure appears to be a hallmark of valid solutions, suggesting a fundamental low-dimensional pattern in the solution space of the graph coloring problem. This can be seen in Figure 2a, providing a visual representation of the model's embedding in 2D. Figure 2b shows the 2D geometry of some message-passing iteration in a failed execution. We will see in the next section that the embedding of both successful and failed execution encode the concepts in a rather similar way, except for the geometry which remains high-dimensional in failed executions.

The analysis demonstrated that the 2D triangular configuration emerged as an invariant pattern across all 100 successful cases, while being notably absent in failed colorings. We

observed that this triangular arrangement manifests during intermediate iterations of the optimization process. Subsequently, the model performs incremental refinements to vertex positions while preserving the triangular topology, ultimately converging to an optimal coloring solution.

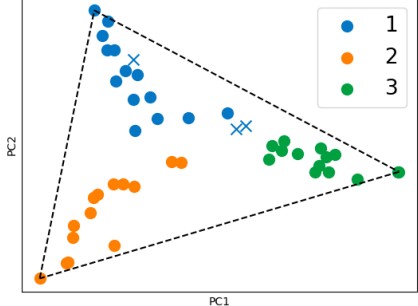

(a) A 2D PCA scatter plot illustrates a valid graph coloring, delineated by a triangular boundary. The plot features three vertices from the graph marked as ×, each with a degree of eight, but different support values: support 4,1,0. The vertex with support 4 is the closest to the blue triangle vertex.

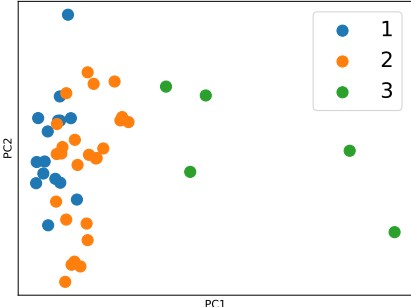

(b) Invalid coloring – triangular shape disappears.

Figure 2: PC1 and PC2 plot of the embedding for successful executions of model 2a, and unsuccessful executions 2b. The successful instance has a triangular shape, unlike the invalid solution.

## 6.2   Distance of neighbors as a confidence mechanism

As the message-passing iterations proceed, we observe that the average Euclidean distance between a vertex's embedding and those of its graph neighbors consistently increases. This phenomenon suggests that the model has discovered the following principle: vertices that should receive different colors are placed far apart in the embedding space. Specifically, for a vertex $v$ with neighborhood $\mathcal{N}(v)$, and embedding $x_v$ we define the confidence as the average neighbor

| Metric | $D_F$ | $D_S$ |
|---|---|---|
| Low-percentile confidence conflicts | 93% ± 10% | 82% ± 23% |
| High-percentile confidence conflicts | 38% ± 15% | 26% ± 15% |
| Spearman between $s(v)$ and $conf(v)$ | 93% ± 12% | 84% ± 13% |
| Spearman between $t$ and $d_v^{(t)}$ | 76% ± 23% | 86% ± 20% |

Table 1: Summary of quantitative concept evaluation on $D_F$ (failed executions) and $D_S$ (successful ones). The values are averages over all executions and iterations (expect for the last metric). "High percentile" is the above the $70^{th}$ percentile, and "Low percentile" is below the $20^{th}$. Successful instances have notable lower values of conflicts between vertices in a given confidence percentile cohort.

| Metric | Sparse ($c < 2.5$) | | Dense ($2.5 \leq c < 5$) | |
|---|---|---|---|---|
| | $D_F$ | $D_S$ | $D_F$ | $D_S$ |
| Low percentile confidence conflicts | 99% ± 3% | 66% ± 27% | 94% ± 6% | 93% ± 5% |
| High percentile confidence conflicts | 44% ± 10% | 32% ± 10% | 31% ± 10% | 24% ± 8% |
| Spearman between $s(v)$ and $conf(v)$ | 97% ± 2% | 88% ± 3% | 93% ± 5% | 88% ± 8% |
| Spearman between $t$ and $d_v^{(t)}$ | 89% ± 21% | 99% ± 0.001% | 67% ± 24% | 76% ± 22% |

Table 2: Breaking down the results in Table 1 by density groups: Sparse ($c < 2.5$) and Dense ($2.5 \leq c < 5$) graphs. Compared to sparse graphs, dense graph show fewer high-percentile conflicts in successful runs and more low-percentile conflicts, aligning with the idea that higher degree increases conflicts for incorrect colorings, but also drives support higher to allow for lower conflict rate at high-confidence quartiles.

distance

$$conf(v) = \frac{1}{|\mathcal{N}(v)|} \sum_{u \in \mathcal{N}(v)} \|x_v - x_u\|_2$$

and find a consistent increase across successive iterations. Quantitative analysis using the average over all executions of the Spearman rank correlation between iteration number and mean neighbor distance yields a strong positive correlation coefficient, with a better score for $D_S$ (86%) than $D_F$ (76%), providing statistical evidence for this progressive spatial separation - See Table 1. Statistical analysis (here and onward) leverages the full set of instances presented in Section 5. This approach, characterized by its unified (on different data distributions) analytical framework, ensures robust statistical validity and mitigates potential distribution-specific biases.

Next, we are going to connect $conf(v)$ and the notion of support $s(v)$. We found that the GNN operates in a way that increases $conf(v)$ more for vertices that also have high values of $s(v)$ (recall that support is the minimum degree among the other two color classes with respect to the current 3-coloring of the graph). To quantify this relation, we measured the Spearman rank correlation between $conf(v)$ and $s(v)$ average over instances, iterations, and vertices and report in Table 1. Both $D_F$ and $D_S$ achieve high spearman ranking, 84% for $D_S$ and 93% for $D_F$. It might seem at first counter-intuitive that failed executions exhibit a better correlation between support and confidence. However, this gap may be explained by the high-dimensional nature of the failed embedding, which allows more freedom, at the expense of the ability to round the high-dimensional solution to a legal discrete solution (the 3-coloring).

Figure 2a further illustrates the pivotal role of the support on top of the degree. Consider three vertices of equal degree

8 but different support: the first vertex has an equal split of neighbors between two color groups ($s(v) = 4$), the second has all but one neighbor in one color group ($s(v) = 1$), and the third has all neighbors in a single color group ($s(v) = 0$). Despite their identical degrees, the vertex with $s(v) = 4$ exhibits higher confidence and is positioned closer to the triangle's vertex. In contrast, the vertices with low support are positioned closer together near the triangle's center, reflecting their lower confidence values. This geometric arrangement demonstrates how the GNN's embedding strategy prioritizes support over degree in determining spatial positioning.

Finally, to find a legal coloring, high confidence, and high support vertices must produce no conflicts along the edges that connect them. Among the set of edges $e = (u, v)$ where both $u$ and $v$ have low confidence and support ( below the $20^{th}$ percentile), there are on average 82% contradictions in $D_S$ (averaged on all instances and iterations) where $D_F$ surfers more contradictions, with an average of 93%. In comparison, the set of edges induced by high confidence and support (both pair vertices have confidence and support above $70^{th}$ percentile) have 26% contradiction on average for $D_S$, and again a higher value for $D_F$, at 38%. The full results are in Table 1.

Thus, we can view the operation of the GNN in two phases: (1) committing to the coloring of the high-confidence-support vertices at an early stage serving as anchors for step (2) where the remaining vertices are placed in the space. Both these operations are guided by iteratively distancing each node from its neighbors.

Finally, let us note that we re-examined the results by density, studying three density groups—Sparse ($c < 2.5$), Dense ($2.5 \leq c < 5$). As evident from Table 2, the trend is preserved - where in successful executions, there are significancy less conflicts in high-percentile confidence than

lower. Notably. in the dense group we found more conflicts in the lower-percentile, both in successful and failed iterations. This makes sense as higher degree leads to more conflicts in case of a wrong coloring assignment. In the high-confidence quartile the trend was reversed, and denser graph had fewer conflicts - again this aligns with the support as proxy for confidence, since higher degree allows for higher support.

## 6.3 The geometry revisited

A surprising fact, at first glance at least, is that the statistics reported in Section 6.2 are quite similar for the failed execution instances in $D_F$ and the successful ones in $D_S$. Therefore, the key to understanding successful vs failed executions lies in the ability to pack the combinatorial properties into a low-dimensional triangular shape. If the embedding is not packed in a 2D triangular shape, then the correlations that we found between distance and support do not translate to reading a legal coloring. In other words, the constrained triangular shape allows the GNN to use the combinatorial information in order to find a low-dimensional representation of the coloring, from which a discrete solution (the 3-coloring) can be usefully read using the $k$-means algorithm. In failed executions, the high-dimensional solution can not be translated well to a discrete one.

The constraint imposed by the triangular geometric shape systematically restricts the latent space movement of vertex embeddings, which elucidates the observed reduction in Spearman rank correlation between support $s(v)$ and conflicts conf$(v)$ in Table 1 for $D_S$. Despite this constraint, the correlation remains substantively high, indicating a nuanced trade-off that fundamentally preserves the method's effectiveness. The reduced dimensional flexibility results in fewer conflicts within the embedding space, ultimately facilitating the discovery of a legal graph coloring.

In optimization theory, the operation of translating high-dimensional vector solutions to discrete solutions is called "rounding". Viewing the operation of GNN-GCP through the lens of its ability to round high-dimensional solution to a legal discrete 3-coloring evokes a striking similarity with the the semi-definite program (SDP) for $k$-coloring. See Appendix A for full details on the SDP.

The objective function of this SDP inherently captures this quest for triangular geometry in its objective function (Goemans and Williamson 1995; Karger, Motwani, and Sudan 1998). For $k = 3$, one optimal solution to the SDP, when the graph is 3-colorable, is derived from the regular 2-simplex (an equilateral triangle). In this case, all vertices in the same color class are assigned vectors pointing from the center of mass of the triangle to one of its vertices. Interpreting the SDP vector solution as an embedding yields a geometry remarkably similar to the one produced by GNN-GCP.

Although the SDP allows for high-dimensional solutions beyond the $k$-simplex geometry (otherwise, the $k$-colorability problem would not have been NP-hard), prior work (Coja-Oghlan, Krivelevich, and Vilenchik 2007) has shown that for some distribution of random 3-colorable graphs, the simplex geometry is the only admissible solu-

tion. Hence, such graphs are easy to color using the SDP algorithm, similar to the GNN-GCP case.

## 7 Conclusion

In this study, we elucidate the principles guiding the GNN-GCP's approach to the 3-coloring problem by combining the geometry of the embedding with the combinatorial concept of support. This integration produces a triangular structure in the embedding space, with high-support vertices serving as anchors for the emerging coloring. Of the various attributes we examined, support and confidence proved to be the most promising and insightful guiding concepts.

Recent research suggests that GNNs can emulate SDP solvers when explicitly designed for that purpose, with their loss functions encoding the SDP objective (Yau et al. 2023). While the GNN-GCP model discussed here was not explicitly trained to act as an SDP solver—it was trained only to decide whether a graph is colorable—it demonstrates behaviors strikingly similar to those of SDP solvers. Despite its open-ended training process and the model not fully learning to utilize these concepts effectively, the GNN-GCP model naturally aligns with heuristic problem-solving strategies grounded in established SDP concepts, which remain the most effective heuristic methods for such problems.

This observation raises several intriguing questions for future research. Does this behavior generalize across other GNN architectures and loss functions for the graph coloring problem? Could similar geometric phenomena emerge in GNN models applied to other combinatorial problems, such as the maximum clique problem? Moreover, can these insights be leveraged to design enhanced versions of GNN-GCP with improved performance, or even inspire the development of more efficient classical algorithms? Penultimately, are there other strategies learned and used in different segments of the iteration? Lastly, Belief Propagation (BP) is a message-passing algorithm that "notifies" each vertex of the probability distribution over the three color classes. Research inspired by statistical mechanics models in physics (Braunstein and Mézard 2005) has also pointed to the notion of support, which anchors the relevant vertices (also called "frozen variables"). Future research can check how BP compares to GNN-GCP and perhaps identify a unifying computational framework tying BP, SDP, and GNNs in the context of combinatorial optimization problems.

## 8 Acknowledgments

We thank the authors of GNN-GCP (Lemos et al. 2019) for sending us their pre-trained weights of the GNN-GCP model- for that, we are grateful.

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

# A  Semi-Definite Program for $k$-Colorability

The $k$-colorability problem for a graph $G = (V, E)$ seeks to assign one of $k$ colors to each vertex such that adjacent vertices receive different colors. This problem can be formulated as a semi-definite program (SDP) by considering its equivalence to the max-$k$-cut problem. Below, we provide the exact SDP formulation and discuss a specific feasible solution based on the $k$-simplex.

## A.1 The Max-$k$-Cut SDP Formulation

The following SDP was introduced by (Goemans and Williamson 1995) as an approximation algorithm for max-$k$-cut, and was then used by (Karger, Motwani, and Sudan 1998) as an approximation algorithm for $k$-coloring. Given a graph $G = (V, E)$, the max-$k$-cut problem is formulated as:

$$\text{maximize} \quad \frac{k}{k-1} \sum_{(i,j) \in E} \left(1 - \mathbf{v}_i \cdot \mathbf{v}_j\right)$$

$$\text{subject to} \quad \mathbf{v}_i \cdot \mathbf{v}_j = -\frac{1}{k-1}, \quad \forall i \neq j,$$

$$\|\mathbf{v}_i\|^2 = 1, \quad \forall i \in V.$$

Here, $\mathbf{v}_i \in \mathbb{R}^n$ represents the vector assigned to vertex $i$, and the constraints ensure that the vectors lie on a $n$-dimensional sphere, $n$ is the number of vertices, with a cosine similarity of $-1/(k-1)$ between two adjacent vertices.

## A.2 The $k$-Simplex Solution

If the graph $G$ is $k$-colorable, then a feasible solution to the SDP is given by embedding the vertices into the vertices of the $k$-dimensional regular simplex. The $k$-simplex is defined as a set of $k$ vectors in $\mathbb{R}^{k-1}$ that are equidistant from each other, satisfying the orthogonality constraint.

Explicitly, the vectors $\mathbf{u}_1, \mathbf{u}_2, \ldots, \mathbf{u}_k$ of the $k$-simplex satisfy:

$$\mathbf{u}_i \cdot \mathbf{u}_j = \begin{cases} 1 & \text{if } i = j, \\ -\frac{1}{k-1} & \text{if } i \neq j. \end{cases}$$

We can assign each vertex $i \in V$ in color class $j$ to the simplex vector $\mathbf{u}_j$, ensuring that adjacent vertices receive distinct simplex vectors, thus satisfying the SDP constraints.

In fact, the SDP solution is also optimal in that case. The objective function is proportional to $(1 - \mathbf{v}_i \cdot \mathbf{v}_j)$, which is maximized, under the constraints, when $\mathbf{v}_i \cdot \mathbf{v}_j = -\frac{1}{k-1}$, as achieved by the $k$-simplex for all $i \neq j$. The factor $\frac{k}{k-1}$ ensures the proper scaling of the objective, making the value of the objective function to be $|E|$, the total number of edges.

While the $k$-simplex is a natural and optimal solution for the SDP, it is not the only possible solution. Other configurations of vectors may also satisfy the constraints of the SDP and yield the same optimal value. For a general graph $G$ this must be the case because the $k$-colorability problem is $\mathcal{NP}$-hard. The SDP is a relaxation of the original problem, and its solution space inherently allows multiple feasible solutions due to its continuous nature.