# OpenReview forum: "From Black Box to Algorithmic Insight: Explainable AI in Graph Neural Networks for Graph Coloring"
_AAAI.org/2025/Workshop/NeurMAD — AAAI 2025 Workshop NeurMAD Submission_

### Official Review · Reviewer_KdjA · 2024-12-20

**Rating:** 7
**Confidence:** 3

**Review:**

Summary:

The paper explores the use of concept learning for XAI in a GNN model trained to solve the 3-coloring problem. It identifies two key concepts, i.e. support and confidence, that are geometrically encoded within the GNN's embeddings, providing interpretable insights into the model's learning process. Although innovative, the study's narrow focus on these concepts restricts its exploration of dynamic embedding evolution, graph topology, and broader applications.

Major concerns:

- The two key concepts analyzed are the support of a vertex and the confidence in the coloring of a specific vertex, evaluated through the 2D PCA projection of node embeddings. This approach appears somewhat limited and may lead to a relatively incomplete exploration of the embedding space. Additional concepts or alternative projections could have been considered for a more comprehensive analysis. Could you please provide insights on this point?
- The paper does not address the different topologies of the graphs or how these topologies might influence the network's learning process. Could you elaborate on this aspect?
- The work discusses the interplay between the static graph structure and dynamic solution states but does not seem to examine how embeddings evolve over time. Additionally, the concepts developed in the study appear to offer a more static representation of the final output generated by the GNN when reaching a solution. A crucial area for further investigation would be to identify specific strategies within the GNN's message-passing framework that dynamically address and resolve color conflicts throughout the solution process. Please add some comments in this.
- Could you please comment on the expected robustness of the introduced concepts compared to typical heuristic methods used for the Graph Coloring Problem?

---

### Decision · Program_Chairs · 2024-12-30

**Decision:**

Accept

**Comment:**

This paper introduces a task that is worth discussing at this workshop. We agree with the reviewer’s opinion.